# GENERATING UNSEEN COMPLEX SCENES: ARE WE THERE YET?

## ABSTRACT

Although recent complex scene conditional generation models generate increasingly appealing scenes, it is very hard to assess which models perform better and why. This is often due to models being trained to fit different data splits, and defining their own experimental setups. In this paper, we propose a methodology to compare complex scene conditional generation models, and provide an in-depth analysis that assesses the ability of each model to (1) fit the training distribution and hence perform well on *seen* conditionings, (2) to generalize to *unseen* conditionings composed of *seen* object combinations, and (3) generalize to *unseen* conditionings composed of *unseen* object combinations. As a result, we observe that recent methods are able to generate recognizable scenes given seen conditionings, and exploit compositionality to generalize to unseen conditionings with seen object combinations. However, all methods suffer from noticeable image quality degradation when asked to generate images from conditionings composed of unseen object combinations. Moreover, through our analysis, we identify the advantages of different pipeline components, and find that (1) encouraging compositionality through instance-wise spatial conditioning normalizations increases robustness to both types of unseen conditionings, (2) using semantically aware losses such as the scene-graph perceptual similarity helps improve some dimensions of the generation process, and (3) enhancing the quality of generated masks and the quality of the individual objects are crucial steps to improve robustness to both types of unseen conditionings.

## 1 INTRODUCTION

The recent years have witnessed significant advances in generative models (Goodfellow et al., 2014; Kingma & Welling, 2014; van den Oord et al., 2016a; Miyato & Koyama, 2018; Miyato et al., 2018; Brock et al., 2019), enabling their increasingly widespread use in many application domains (van den Oord et al., 2016b; Vondrick et al., 2016; Zhang et al., 2018a; Hong et al., 2018; Sun & Wu, 2020). Among the most promising approaches, Generative Adversarial Networks (GANs) (Goodfellow et al., 2014) have achieved remarkable results, generating high quality, high resolution samples in the context of *single class conditional* image generation (Brock et al., 2019). This outstanding progress has paved the road towards tackling more challenging tasks such as the one of *complex scene conditional* generation, where the goal is to generate high quality images with multiple objects and their interactions from a given conditioning (e.g. bounding box layout, segmentation mask, or scene graph). Given the exploding number of possible object combinations and their layouts, the requested conditionings oftentimes require zero-shot generalization. Therefore, successfully generating high quality, high resolution, diverse samples from complex scene datasets such as COCO-Stuff (Caesar et al., 2018) remains a stretch goal.

Despite recent efforts producing increasingly appealing complex scene samples (Hong et al., 2018; Hinz et al., 2019; Park et al., 2019; Ashual & Wolf, 2019; Sun & Wu, 2020; Sylvain et al., 2020), and as previously noted in the *unconditional* GAN literature (Lucic et al., 2018; Kurach et al., 2018), it is unfortunately very hard to assess which models perform better, and perhaps more importantly why. In the case of conditional complex scene generation, this is often due to models being trained to fit different data splits, using different conditioning modalities and levels of supervision – bounding box layouts, segmentation masks, scene graphs –, and reporting inconsistent quantitative metrics (e.g. repeatedly computing previous methods' results using different reference distributions, and/or

using different image compression algorithms to store generated images), among other uncontrolled sources of variation. Moreover, these methods disregard the challenges that emerge from their expected generalization to unseen conditionings. This lack of rigour leads to conclusion replication failure, and hinders the identification of the most promising directions to advance the field.

Therefore, in this paper we aim to provide an in-depth analysis and propose a methodology to compare current conditional complex scene generation methods. We argue that such analysis has the potential to deepen the understanding of such approaches and contribute to their progress. The proposed methodology addresses the following questions: (1) How well does each method perform on *seen* conditionings (training conditionings)?; (2) How well does each method generalize to *unseen* conditionings composed of *seen* object combinations?; and (3) How well does each method generalize to *unseen* conditionings composed of *unseen* object combinations? We investigate the answers to the previous questions both from a scene-wise and an object-wise standpoints. As a result, we observe that: (1) recent methods are capable of generating identifiable scenes from seen conditionings; (2) they exhibit some generalization capabilities when using unseen conditionings composed of seen object combinations, exploiting compositionality to generate scenes with different object arrangements; (3) they suffer from poorer generated image quality when asked for unseen object combinations, especially the method presented in (Sylvain et al., 2020). However, in all cases, the quality of the generated objects generally suffers from missing high frequency details, especially for those classes in the long tail of the dataset distribution. Finally, through an extensive ablation, we are able to identify the strengths and weaknesses of different pipeline components. In particular, we note that endowing the generator with an instance-wise normalization module (Sun & Wu, 2019) results in increased individual object quality and better robustness to both types of unseen conditionings, whereas exploiting the normalization module of Sylvain et al. (2020) results in improved scene quality, suggesting that exploiting scene compositionality in the generator helps improve generalization. Moreover, we find that including a scene-graph similarity (Sylvain et al., 2020) while training complex scene conditional generation models leads to better conditional consistency, especially for unseen object combinations, emphasizing the promise of moving towards semantically aware training losses to improve generalization. We also identify the improvement of generated segmentation masks as one promising avenue to promote generalization to unseen conditionings. Finally, by leveraging these findings, we are able to compose a pipeline which obtains state-of-the-art results in metrics such as FID, while maintaining competitive results in almost all other studied metrics.

## 2 RELATED WORK

**Evaluation metrics for GANs**. The most widely used metrics are the Inception Score (IS) (Salimans et al. (2016)) and the Fréchet Inception Distance (FID) (Heusel et al. (2017)), which aim to capture visual sample quality and sample diversity. On the one hand, IS was designed to evaluate single-object image generation on problems where the expected marginal distribution over classes is uniform, an unrealistic expectation in many real-world datasets with multi-object images. On the other hand, FID was introduced for unconditional image generation and yields a single distribution-specific score, which hinders the analysis of individual failure cases. To overcome the former, researchers have attempted to extend FID to the conditional generation case (DeVries et al., 2019; Benny et al., 2020). To overcome the latter, researchers have proposed to disentangle visual sample quality and diversity into two different metrics (Shmelkov et al., 2018; Sajjadi et al., 2018; Kynkäänniemi et al., 2019; Ravuri & Vinyals, 2019) by modifying the definition of precision-recall in different ways. Finally, the diversity of generated samples has also been quantified by measuring the perceptual similarity between generated image patches, as proposed by Zhang et al. (2018b).

**Complex scene conditional generation**. The conditional generation literature encompasses a variety of input conditioning modalities. Several existing works focus on generating photo-realistic scenes from detailed semantic segmentation masks (Chen & Koltun, 2017; Qi et al., 2018; Park et al., 2019; Wang et al., 2018; Tang et al., 2020). Although semantic segmentation masks are information rich, they may be difficult to obtain with enough fine-grained details. This is especially relevant in applications where a user is expected to specify the input conditioning. As such, other commonly used input conditioning modalities include text descriptions (Reed et al., 2016; Hong et al., 2018; Tan et al., 2019), scene graphs (Johnson et al., 2018; Ashual & Wolf, 2019), and bounding box layouts specifying the position and scale of the objects in the scene (Zhao et al., 2019; Sun &

Wu, 2019; 2020; Sylvain et al., 2020). In addition to those, recent works have focused on generating bounding box layouts from a set of classes (Jyothi et al., 2019) and designing remarkably flexible pipelines to enable user interactions (Ashual & Wolf, 2019). From a model perspective, complex scene conditional generation has been improved through the development of tailored class-aware (Park et al., 2019), instance-aware (Sun & Wu, 2019; 2020), and class-and-instance-aware normalizations (Sylvain et al., 2020) incorporated in the GAN generator. The task has also benefited from the introduction of refinement pipelines (Sun & Wu, 2020), as well as the design of a multi-modal perceptual loss to drive scene graph embeddings and generated image embeddings close to each other (Sylvain et al., 2020). All aforementioned models are often trained on a different number of training samples, use different backbone architectures, and report metrics on different splits and/or using different pre-trained models. This results in repeated reevaluations of baselines to fit each methods' setup, making it hard to pinpoint the most promising research directions.

## 3 METHODOLOGY

We start by defining three sets of data, composed of triplets of images and their respective *finegrained* and *coarse* conditionings. The finegrained conditionings represent detailed scene layouts by means of scene graphs, bounding boxes or segmentation masks. The coarse conditionings are an abstraction of the finegrained ones into a set of object labels present in the scene. For example, given a scene layout depicting two glasses on a table, the corresponding set of labels is {*glass*, *table*}.

Let $\mathcal{D}_s$ be a set containing triplets of real images and their corresponding conditionings. We use $\mathcal{D}_s$ to train the complex scene conditional generation models, and hereinafter refer to the conditionings used to train the models as *seen* conditionings. Let $\mathcal{D}_u$ be a set containing triplets of real images and their corresponding conditionings, composed of *seen* coarse conditionings but *unseen* finegrained conditionings. Note that we use the term *unseen* to refer to conditionings that are not present in $\mathcal{D}_s$. Similarly, let $\mathcal{D}_{u^2}$ be a set containing triplets of real images and their corresponding conditionings, composed of *unseen* coarse conditionings and *unseen* finegrained conditionings. We use $\mathcal{D}_u$ and $\mathcal{D}_{u^2}$ only for evaluation purposes. After training the models, we use the finegrained conditionings in $\mathcal{D}_s$, $\mathcal{D}_u$ and $\mathcal{D}_{u^2}$ to generate three sets of model samples: $\mathcal{S}_s$, $\mathcal{S}_u$ and $\mathcal{S}_{u^2}$.

The goal of our analysis is three-fold: (1) to assess how well each model fits the training data, (2) to assess each models' generalization to unseen finegrained conditionings, and (3) to assess each models' generalization to unseen coarse conditionings. To do so, we measure discrepancies between the distributions defined by the images in $\mathcal{D}_s$, $\mathcal{D}_u$ and $\mathcal{D}_{u^2}$ and by the images in $\mathcal{S}_s$, $\mathcal{S}_u$ and $\mathcal{S}_{u^2}$, respectively. In particular, we evaluate the quality of the generation processes from both *scene-wise* and *object-wise* standpoints. Note that the scene-wise evaluation considers full real images and full generated images, whereas the object-wise evaluation considers crops of individual objects from real and generated images. Throughout our analysis, we consider the following metrics to assess scene-wise and object-wise generation processes: precision, recall, consistency, FID, LPIPS-based diversity score (DS), and object classification accuracy or image(scene)-to-set prediction F1-score. In the remainder of this section, we will use *image* to refer to either full images or cropped objects.

We follow the definition of precision and recall introduced by Kynkäänniemi et al. (2019). Intuitively, precision captures the sample quality, whereas recall captures the coverage of the generated distribution. In particular, precision verifies whether generated images lie within a reference manifold of real images, and recall verifies whether real images lie within a reference manifold of generated images. The real and generated manifolds are estimated in the feature space, which in our case is defined by a ResNext101 (Xie et al., 2017), and are obtained surrounding each data point with a hyper-sphere that reaches its $k$-th ($k = 5$) nearest neighbor. Therefore, precision and recall are sensitive to the number of samples used to estimate the reference manifolds, resulting in overestimated manifold volumes when the sample size is small. Hence, when $|\mathcal{D}_u|$ and $|\mathcal{D}_{u^2}|$ are small, we estimate the manifold's hyper-sphere radii using the images in $\mathcal{D}_s \cup \mathcal{D}_u \cup \mathcal{D}_u^2$. Analogously, we estimate the manifold of generated images from $\mathcal{S}_s \cup \mathcal{S}_u \cup \mathcal{S}_u^2$. Although successful conditional generation models should respect their conditionings and generate high quality samples which are consistent with those, the adopted definition of precision does not guarantee that the reference real sample and the generated sample found in its hyper-sphere will have the same coarse conditionings. Therefore, we complement precision with a metric to measure the consistency of the generated samples that fall within the estimated real manifold. More precisely, we compute the intersection over union

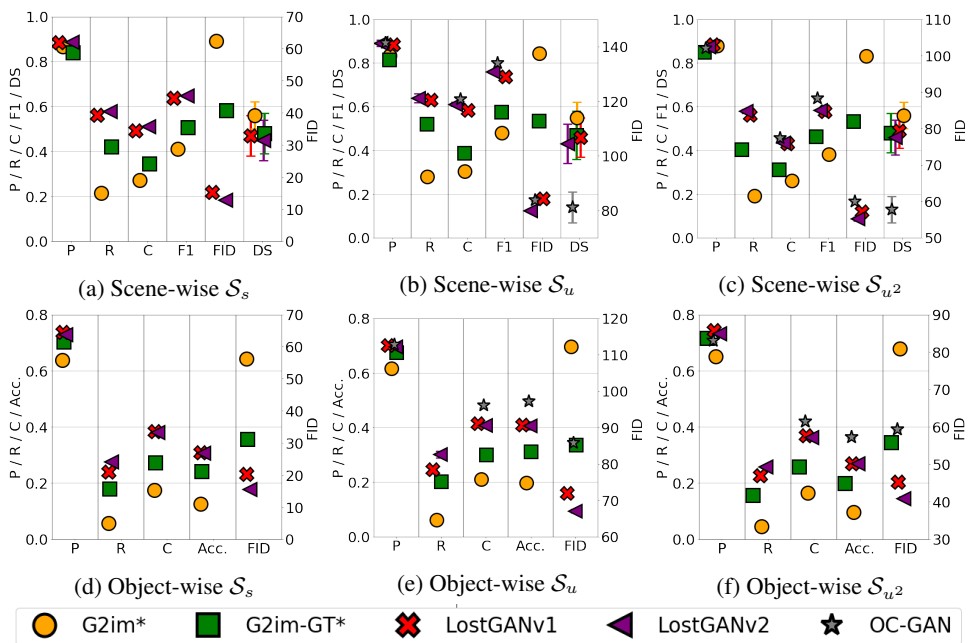

Figure 1: Comparison of state-of-the-art methods in terms of precision (P), recall (R), consistency (C), F1-score (F1), object accuracy (Acc.), FID and DS for (a-d) seen conditionings ($\mathcal{S}_s$), (b-e) unseen finegrained conditionings ($\mathcal{S}_u$), (c-f) unseen coarse conditionings ($\mathcal{S}_{u^2}$). Plots show mean and std over 5 generation processes. *Trained using the open-sourced code. Best viewed in color.

(IoU) between the classes present in the reference sample and those present in the conditioning of the generated sample. The consistency is averaged over all generated images, assigning consistency 0 to samples lying outside the estimated manifold.

## 4 ANALYSIS

We perform our analysis using five recently introduced complex scene generation pipelines: (1) **G2im** (Ashual & Wolf, 2019); (2) **G2im-GT** (Ashual & Wolf, 2019) which is a variant of G2im, where the instance segmentation masks and instance bounding box layouts are not generated, but directly taken from the ground-truth data; (3) **LostGANv1** (Sun & Wu, 2019); (4) **LostGANv2** (Sun & Wu, 2020) which is an improved version of LostGANv1; and (5) **OC-GAN** (Sylvain et al., 2020). Detailed description of these approaches can be found in the Appendix C.

### 4.1 DATASET

We perform our analysis on the challenging COCO-Stuff (Caesar et al. (2018)), given its ubiquitous use in conditional image generation. The training and evaluation splits consist of 75k and 3k images respectively, following Sun & Wu (2019). We use the whole training set as $\mathcal{D}_s$, and divide the validation set into three sets, namely $\mathcal{D}_u$, $\mathcal{D}_{u^2}$ and a set used for hyper-parameter search and validation $\mathcal{D}_v$[1]. Note that $\mathcal{D}_v$ is defined and used as in (Ashual & Wolf, 2019). We use 128x128 resolution images, as all studied methods support this resolution. Finally, it is worth noting that the proportions of long-tail classes[2] in the previously defined $\mathcal{D}_s$, $\mathcal{D}_u$ and $\mathcal{D}_{u^2}$ are 47%, 37% and 50%, respectively, resulting in more well represented classes in $\mathcal{D}_u$ than $\mathcal{D}_{u^2}$.

---

[1]$\mathcal{D}_v$ is included when computing the reference manifold, and its corresponding generated images $\mathcal{S}_v$ are included when computing the manifold of generated images.

[2]We consider long-tail classes the non top-25 most frequent classes in the COCO-Stuff dataset.

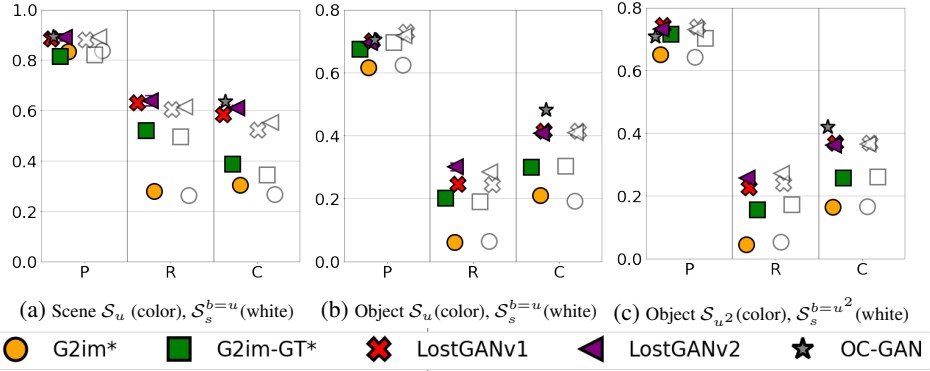

(a) Scene $\mathcal{S}_u$ (color), $\mathcal{S}_s^{b=u}$(white)  (b) Object $\mathcal{S}_u$(color), $\mathcal{S}_s^{b=u}$(white)  (c) Object $\mathcal{S}_{u^2}$(color), $\mathcal{S}_s^{b=u^2}$(white)

Figure 2: Comparison of state-of-the-art methods in terms of precision (P), recall (R), and consistency (C), and matching class distributions for (a-b) seen and unseen finegrained conditionings, (c) seen and unseen coarse conditionings. *Trained using the open-sourced code. Best viewed in color.

## 4.2 FITTING SEEN CONDITIONINGS

We start by analyzing the quality of the generation processes of the methods from a scene-wise perspective[3]. All methods have been trained for 125 epochs to give them equal opportunity to fit the training data. Figure 1a depicts the precision (P), recall (R), consistency (C), F1-score (F1), FID and DS achieved by the methods under study. We observe that all methods have rather high precision scores ($\sim 90\%$). Despite its remarkable precision, G2Im and G2Im-GT suffer from poor recall and consistency. At the same time, LostGANv1 and LostGANv2 exhibit at least $\sim 10\%$ higher recall and consistency than G2Im and/or G2Im-GT, with LostGANv2 being slightly better than LostGANv1, hence suggesting better generated distribution coverage and conditional consistency, which could be attributed to its mask refinement process and/or its larger number of parameters. Despite the superiority of LostGANs, their achieved consistency is only around 50%, indicating that image conditionings are often violated. Additionally, their recall is slightly below 60%, which implies that about 40% of the real images fall outside of the manifold of generated images. Moreover, when compared to Grid2Im-GT, LostGANs' superiority cannot be attributed to having access to improved layouts (Grid2Im-GT uses ground truth layouts), but rather to their modeling choices. It is worth noting that although G2Im has higher precision than G2Im-GT, G2Im-GT obtains significantly better recall. This could suggest that Grid2Im is only able to model some modes of the layout distribution. If we consider F1-score and FID, we see that the trend is preserved, with LostGANs being the top performers. When it comes to DS, all methods behave similarly. Although recall can also be thought of as a diversity metric, it is concerned with diversity in the whole image space, whereas DS tries to capture the intra-conditioning diversity. The absence of a clear leading method in terms of DS raises the question whether there could be any major improvement given the nature of the dataset, which contains a single instantiation of each finegrained conditioning.

Figure 1d shows a similar trend for object-wise metrics, with LostGANv2 scoring higher recall but similar precision and consistency than LostGANv1, suggesting that refined masks improve more scene than object quality. It is worth noting that object-wise metrics exhibit lower scores than scene-wise metrics, suggesting that the generation processes might lead to appealing generated scenes, which however suffer from high frequency details.

Finally, Figure 3 depicts two examples of images resulting from conditional generation processes from seen conditionings, showing in most cases recognizable objects in the scene, which could however be significantly improved, and emphasizing the quantitative superiority of LostGANv2.

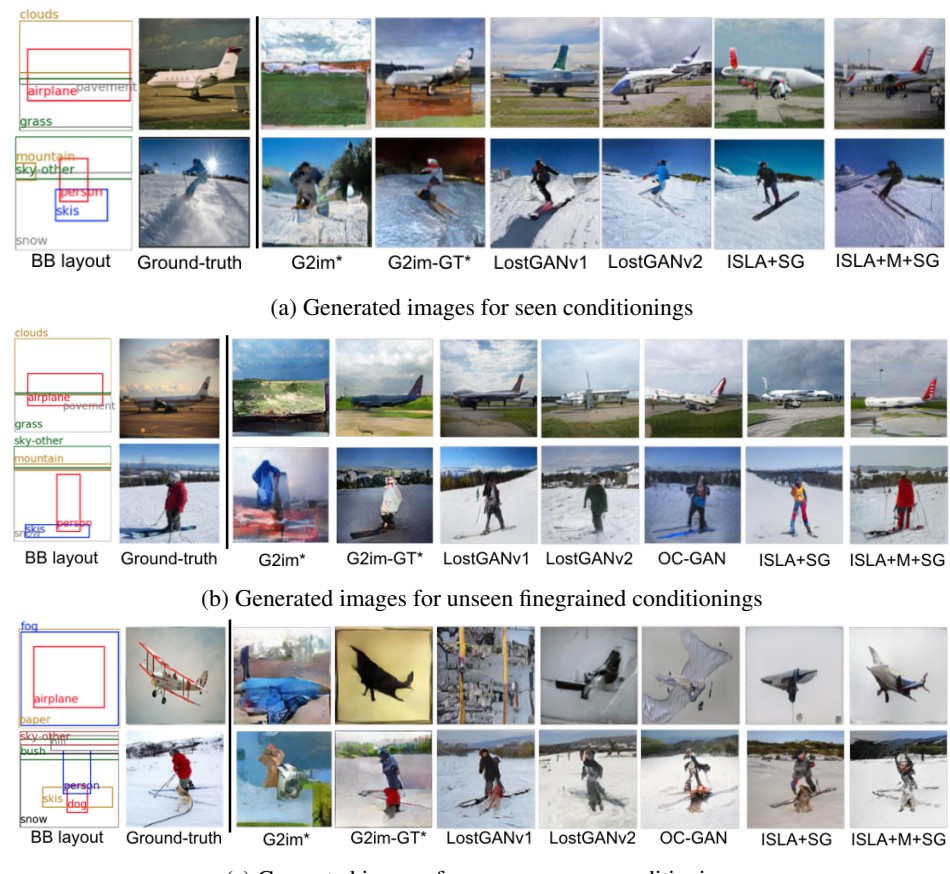

(a) Generated images for seen conditionings

(b) Generated images for unseen finegrained conditionings

(c) Generated images for unseen coarse conditionings

Figure 3: Qualitative results. Generated images with resolution $128 \times 128$ for several methods.

## 4.3 GENERALIZATION TO UNSEEN FINEGRAINED CONDITIONINGS

Here we compare the quality of the generation processes of all methods, when generating images from unseen finegrained conditionings ($\mathcal{S}_u$). We start by assessing the methods' performances from both scene-wise and object-wise standpoints, see Figures 1b and 1e. In this case, we consistently see that OC-GAN and LostGANv1/v2 show the most competitive results across most of the metrics. In particular, OC-GAN performs on par with the best version of LostGAN for both scene and object-wise precision, i.e. it matches the top scene precision performer and the top object precision performer at the same time, suggesting that this method does not trade scene precision for object precision. Moreover, OC-GAN consistently exhibits improved performance in terms of scene/object consistency, as well as scene F1-score and object accuracy, which highlights that its high scene/object precision also respects the (coarse) conditionings. When it comes to FID, Lost-GANv2 leads both the scene-wise and object-wise rankings, reiterating their ability to generate high quality images from unseen finegrained conditionings. However, OC-GAN's object-wise FID ranks the model as $3rd$, together with G2Im-GT, whereas G2Im-GT achieves worse FID than OC-GAN scene-wise. This suggests that OC-GAN may suffer from poorer object diversity, resulting in lower FID, and that having access to high quality (GT) layouts may lead to better object quality. When it comes to DS, OC-GAN has by far the lowest result, suggesting the method trades image quality for intra-conditioning diversity. G2Im and G2Im-GT are the weakest performers, with G2Im-GT achieving more competitive results.

---

[3]OC-GAN's code is not publicly available, nor are its pretrained models. The authors kindly shared their evaluation images $\mathcal{S}_u$ and $\mathcal{S}_{u^2}$ through personal communication. Those were obtained after a 170 epochs training process. Unfortunately, we could not get any generated images from the training conditionings, and so we do not consider this method to evaluate $\mathcal{S}_s$ nor do we compute its recall in the next sections.

| | ↑SP | ↑SR | ↑SC | ↑OP | ↑OR | ↑OC | ↑F1 | ↑ Acc. | ↑DS | ↓SFID | ↓OFID |
|---|---|---|---|---|---|---|---|---|---|---|---|
| *unseen finegrained conditionings $\mathcal{S}_u$* | | | | | | | | | | | |
| ISLA | 89.7 | **65.2** | 63.6 | 73.1 | 28.4 | 45.9 | 76.7 | 43.1 | **0.41** | 78.5 | 66.1 |
| SPADE(oc) | 90.9 | 57.0 | 65.7 | 70.2 | 26.6 | 41.5 | 75.9 | 39.3 | 0.17 | 78.7 | 67.6 |
| ISLA + M | 91.6 | 57.7 | 68.1 | **73.8** | 31.6 | 49.3 | 77.0 | 45.4 | 0.12 | 75.2 | 61.7 |
| ISLA + SG | 90.5 | 64.9 | 66.1 | 70.5 | **32.2** | 45.3 | 78.9 | 42.1 | **0.41** | 78.4 | 62.6 |
| ISLA + M + SG | **92.1** | 58.8 | **72.5** | 72.6 | 28.7 | **50.6** | **80.8** | 49.3 | 0.18 | **72.6** | **59.2** |
| *unseen coarse conditionings $\mathcal{S}_{u^2}$* | | | | | | | | | | | |
| ISLA | 88.4 | 58.5 | 45.9 | 74.7 | 24.9 | 40.5 | 59.6 | 30.7 | **0.44** | 53.5 | 38.2 |
| SPADE(oc) | 90.1 | 53.2 | 47.6 | 73.6 | 24.5 | 35.7 | 59.2 | 26.4 | 0.20 | 55.6 | 40.2 |
| ISLA + M | 91.7 | 55.3 | 53.1 | **77.7** | 28.1 | 43.4 | 61.1 | 29.0 | 0.11 | 52.3 | 36.0 |
| ISLA + SG | 88.4 | **59.1** | 47.2 | 74.2 | **28.3** | 42.7 | 62.2 | 31.4 | **0.44** | 52.1 | 34.2 |
| ISLA + M + SG | **91.8** | 54.8 | **55.9** | 74.3 | 25.0 | **43.7** | **64.3** | **34.5** | 0.19 | **50.2** | **33.2** |

Table 1: Ablation results. We report average precision, recall and consistency for scenes (SP, SR, SC) and objects (OP, OR, OC), F1-score, Accuracy (Acc.), DS, scene and object FIDs (SFID and OFID). Results show the mean over 5 random seeds at test time. Extended table in Appendix E.

When analyzing the generalization gap, we see that scene-wise precision achieves the same score for both the seen conditionings ($\mathcal{S}_s$) and the unseen finegrained ones ($\mathcal{S}_u$), whereas object-wise precision is slightly better for $\mathcal{S}_s$ than for $\mathcal{S}_u$. However, recall and consistency show consistently better scores for $\mathcal{S}_u$ than for $\mathcal{S}_s$ (see Figures 1a, 1b, 1d, 1e). We hypothesize that the lower presence of long-tail classes in $\mathcal{S}_u$ results in better metrics for this split. To confirm the hypothesis, we subsample $\mathcal{D}_s$ to match the same class distribution as in $\mathcal{D}_u$, creating the split $\mathcal{D}_s^{b=u}$, and generate the corresponding set of model samples $\mathcal{S}_s^{b=u}$. Results are presented in Figure 2, and show that for $\mathcal{S}_s^{b=u}$ the recall performance gap with $\mathcal{S}_u$ is roughly closed, whereas the consistency performance gap becomes smaller. Overall, the results observed for $\mathcal{S}_u$ indicate good generalization to unseen finegrained conditionings for all methods. This is further supported by the visualizations in Figure 3, where the quality of generated images for unseen finegrained conditionings equates the one of generated images for seen conditionings, suggesting that state-of-the-art models properly exploit the compositionality of the scenes, and as such are able to generate images with recognizable objects from unseen finegrained conditionings.

## 4.4 GENERALIZATION TO UNSEEN COARSE CONDITIONINGS

Here we compare the quality of the generation processes for all methods, when generating images from unseen coarse conditionings ($\mathcal{S}_{u^2}$). We start by assessing the methods' performances from both scene-wise (see Figure 1c) and object-wise (see Figure 1f) standpoints. Similarly to previous experiments, we observe that all methods achieve high precision values, especially scene-wise, and relatively low recall and consistency values. G2Im exhibits competitive results in terms of scene average precision but lags behind in the vast majority of the metrics. Interestingly, the high scene precision of G2Im does not translate to its object precision, suggesting that the method does not capture enough high frequency detail. The overall quality of the scene generation process of OC-GAN is mostly on par with the one of LostGANv1/v2, achieving similar precision and consistency and slightly lower scene FID score. However, OC-GAN results in lower FID scores object-wise, suggesting that OC-GAN suffers from object diversity, as in $\mathcal{S}_u$. Once again, OC-GAN shines when it comes to preserving conditional consistency (see consistency, F1, object accuracy metrics) and but appears to be the weakest in terms of DS.

Interestingly, when analyzing the generalization gap, we see that scene-wise and object-wise precision and recall achieve roughly the same scores for the seen conditionings ($\mathcal{S}_s$) and the unseen coarse ($\mathcal{S}_{u^2}$) ones. However, scene and object consistencies are more accurate for $\mathcal{S}_s$ than for $\mathcal{S}_{u^2}$. If we compare the generalization to $\mathcal{S}_u$ against the generalization to $\mathcal{S}_{u^2}$, we observe that for scenes and objects, $\mathcal{S}_{u^2}$ presents lower recall and consistency for most of the methods. Further, object-wise precision is also affected for $\mathcal{S}_{u^2}$, altering the methods' ranking. We note that OC-GAN loses its $1st$ ranked position (shared with the LostGANs) in $\mathcal{S}_u$ to become $3rd$ (together with G2Im-GT) in $\mathcal{S}_{u^2}$, and suffers an important loss in terms of object FID, suggesting that OC-GAN may suffer particularly from unseen coarse conditionings. For completeness, and to assess whether the marginally lower recall and consistency of $\mathcal{S}_{u^2}$ come from the heavier long tail of $\mathcal{S}_{u^2}$, we subsample $\mathcal{D}_s$ to

match the same class distribution as in $\mathcal{D}_{u^2}$. In this case, this experiment can only be performed object-wise, as none of the class combinations in $\mathcal{D}_{u^2}$ appears in $\mathcal{D}_s$. The results presented in Figure 2 show that precision, recall and consistency of both splits are very similar.

Finally, looking at Figure 3, we observe that images generated from unseen coarse conditionings present lower quality than images generated from seen or unseen finegrained conditionings for all methods. For instance, if we consider the conditioning in Figure 3c, which depicts a person skiing with a dog, we observe that the overall quality of the generated images is rather low, with oversmoothed or implausible looking persons and, in some cases, invisible dogs. Similarly, if we consider the conditioning of an airplane on a paper, most of the resulting generated images also exhibit rather low quality. The most compelling image is the one generated by G2Im-GT, which directly leverages a high quality (GT) layout and can focus on filling in the object textures. Additional qualitative results can be found in Appendix F.

### 4.5 Ablation of pipeline components

In this subsection, we switch the gears of the analysis and design an ablation study to better understand how elements of complex scene generation pipelines affect their performance. In particular, we inspect components that were either presented as a novelty in the methods under study or that are an important defining factor of these approaches. More precisely, we analyze the following components: (1) the spatial conditioning normalization module; (2) the effect of imperfect instance segmentation mask prediction on the image quality; and (3) the influence of scene graph loss introduced in OC-GAN. We carry our analysis on models derived from LostGANv2. Note that the multi-stage generation process introduced in (Sun & Wu, 2020) constitutes the fundamental difference between LostGANv1 and LostGANv2 and, as such, is already under study in Subsections 4.2–4.4. We use hyperbands (Li et al., 2017) to find the best hyperparameters for each model.

**Spatial conditioning normalization modules**. We compare LostGANv2's ISLA-norm to the variant of SPADE introduced in OC-GAN, hereinafter called SPADE(oc). The results of these comparisons are presented in Table 1 both for unseen finegrained conditionings and for unseen coarse conditionings. When comparing ISLA to SPADE(oc) in the unseen finegrained conditionings scenario, we observe that ISLA improves upon diversity metrics such as scene recall, object recall and DS on average. The high scene variation achieved by ISLA could possibly be explained given it design, which leverages latent vectors to compute instance-wise normalizing parameters. Interestingly, ISLA also favors object-wise metrics, by improving object consistency and accuracy. The same trend can be observed in the case of unseen coarse conditionings. This might be due to the fact that ISLA considers individual objects information through its input. However, SPADE(oc) dominates the scene-wise metrics, leading to better scene precision and scene consistency on average. This might be due to the module's design, which directly exploits bounding box semantic segmentation masks. These results suggest that exploiting scene compositionality in the generator's spatial conditionings (as in ISLA) helps improve generalization. Additional ablations with variants of SPADE can be found in Appendix E.

**Ground-truth masks (M)**. As expected, using GT masks in ISLA+M greatly improves over ISLA. In particular, for $\mathcal{S}_u$, using ground truth masks improves upon most of the ISLA results on average, at the expense of scene recall and DS decrease. We observe the same trend for $\mathcal{S}_{u^2}$, where improvements are mainly observed in terms of object and scene precision and consistency metrics. Again, using GT masks reduces the scene layout variability and translates into lower diversity-related scores. Moreover, using unseen GT masks forces the generator to hallucinate objects with unseen shapes. This may in some cases lead to undesirable outcomes, with worse generated image quality than when using predicted masks. However, generating good quality masks remains one of the most promising directions to improve generalization.

**Scene-graph similarity module (SG)**. When endowing ISLA with SG (ISLA+SG), we observe notable boosts in scene-wise conditional consistency metrics in $\mathcal{S}_u$ (see scene consistency and F1). This is not surprising given that the module's goal is to match information from scene graphs and generated images to encode the location of specific objects in the scene. It is worth noting that adding the SG module does not come at the expense of intra-conditioning diversity but harms scene recall on average. In $\mathcal{S}_{u^2}$, we observe a marked increase in object recall, object consistency, F1, and both FIDs, while noticing a slight decrease in terms of average precision, emphasizing the general

benefits of the SG module for unseen coarse conditionings. When using both GT masks and the SG module (ISLA+M+SG), we observe further improvements. For both $\mathcal{S}_u$ and $\mathcal{S}_{u^2}$, ISLA+M+SG achieves the highest results in terms of average scene precision, consistency, F1, as well as FID, and improves upon object consistency, accuracy and FID, while harming diversity metrics. Note that ISLA+M+SG slightly mitigates the decrease in DS/recall experienced by ISLA+M. The compelling improvements of ISLA+M+SG suggest that the quality of the generated masks plays a crucial role in improving generalization to unseen conditionings. Figure 3 shows generated images for both variants of the model with the SG module. We can see that the generated images display high frequency details for the scenarios of seen and unseen finegrained conditionings. For unseen coarse conditionings, we can appreciate better results for ISLA+SG that could be combining both instance-wise compositionality from ISLA and the consistency boost of SG module. Moreover, ISLA+M+SG leverages GT masks, showing further quality improvement. The advantages of using SG suggest that semantically aware training losses promote generalization and opens up future research avenues in this direction.

Finally, as a byproduct of our ablation, we were able to come up with model modifications that lead to superior generalization, achieving state-of-the-art results with ISLA+SG. In particular, we outperform LostGANv2 – denoted as ISLA in Table 1 –, which was the best method in terms of FID to the date. More precisely, we achieve lower scores for both scene and object FID and for both generalization scenarios ($\mathcal{S}_u$ and $\mathcal{S}_{u^2}$), while matching or surpassing for the other reported metrics. For a more detailed comparison with state-of-the-art methods, refer to Appendix H.

## 5 CONCLUSIONS

We proposed a methodology to analyze current complex scene conditional generation methods. Using this methodology, we evaluated recent complex scene generation pipelines from the points of view of training data fitting as well as their ability to generalize unseen scenes. Through an extensive evaluation, we observed that current methods: (1) fit the training distribution with a moderate success, generating recognizable scenes which can however still be notably improved to enhance the overall quality of the generation process; (2) display decent generalization to unseen finegrained conditionings; (3) have significant space for improvement when it comes to generating images from unseen coarse conditionings; and (4) the individual quality of the generated objects generally suffers from missing high frequency details, especially for objects belonging to the long tail classes of the dataset. Thus, complex scene generation is still a stretch goal, especially for unseen conditionings, where the methods' performances are still in their infancy. To provide some pointers on how to move towards high quality complex scenes generation for unseen scenarios, we performed an ablation study that highlights the importance of model components when it comes to generalization, and show that by exploiting our findings we are able to achieve a new state-of-the-art in most reported metrics. We note that promising avenues to improve generalization in complex scene conditional generation include: (1) incorporating compositional priors in the generator; (2) moving towards semantically aware training losses; and (3) focusing on improving mask generation as a way to improve scene generation quality.

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

## A    EXTENDED EXPERIMENTAL SETUP

**Dataset details**. We use COCO-Stuff (Caesar et al. (2018)), with training and evaluation splits consisting of 75k and 3k images respectively, following Sun & Wu (2019). We use the whole training set as $\mathcal{D}_s$, and divide the validation set into three sets, namely $\mathcal{D}_u$, $\mathcal{D}_{u^2}$ and a set used for hyper-parameter search and validation $\mathcal{D}_v$. This results in $|\mathcal{D}_s| = 75$k, $|\mathcal{D}_u| = 675$, $|\mathcal{D}_{u^2}| = 1375$ and $|\mathcal{D}_v| = 1024$ triplets to evaluate the quality of the generation processes. We use the above described splits to compute the scene-wise metrics, and crop individual objects from the scenes to compute object-wise metrics. In particular, we consider objects from *things* categories exclusively, given that crops of *stuff* categories most often also contain foreground objects from *things* categories. This results in 145k objects crops extracted from $\mathcal{D}_s$, 1163 from $\mathcal{D}_u$, 2706 from $\mathcal{D}_{u^2}$ and 2062 from $\mathcal{D}_v$. In both cases and only for evaluation purposes, we preprocessed the data sets by merging redundant and prone-to-confusion classes (see Appendix B for a detailed explanation of the dataset preprocessing).

**Evaluation networks**. Our object classifier has a ResNext101 backbone and is trained on individual objects cropped from the images in $\mathcal{D}_s$. Similarly, the image(scene)-to-set predictor has a ResNext101 backbone and is trained on $\mathcal{D}_s$. In both cases, the architecture and optimization hyper-parameters are chosen independently on $\mathcal{D}_v$ and by means of the Hyperband algorithm (Li et al., 2017), following (Pineda et al., 2019). Both evaluation networks are trained on the merged COCO-Stuff categories.

**Evaluation details**. We report all results with 128x128 images, as all studied methods support this resolution size. The image features to compute precision, recall and consistency metrics are extracted from the evaluation networks (image-to-set or object classifier). In both cases, we use the features from the last fully connected layer before the classifier. For each finegrained conditioning, we generate images with each method from 5 random seeds, which change the input noise $z$ fed to the generators, and so allow for some image variation. The metrics are computed and averaged across the 5 seeds for $\mathcal{S}_s$, $\mathcal{S}_u$ and $\mathcal{S}_{u^2}$.

## B    DATASET DETAILS AND CATEGORIZATION CLEANING

In COCO-Stuff the distinctions between *stuff* categories and some thing categories is difficult to grasp even for humans. Analyzing a confusion matrix extracted for feature extractor on object crops and some designed rules, we cleaned the COCO-Stuff category list to minimize mistakes in evaluation due to poor dataset categorization. Note that is is only done for evaluation, since all studied generative models have been trained with the original COCO-Stuff categories.

We extracted VGG16 Simonyan & Zisserman (2014) features for all cropped training objects. We computed pair-wise Euclidean distances between all feature crops, and assigned as the predicted class the one corresponding to the closest crop. Using these class assignments, we obtain a confusion matrix, that illustrates which are the most confused class and by what other classes they are confused. To decide which classes to merge, we applied the following steps:

- For each target class class, we took the 5-top most confused classes that have, at least, the same probability as the target class.

- We discarded the class 'person' from any possible confusion, as this miss-classification can come from the bounding box crop.
- We also do not consider some confused classes where the confusion can come from the bounding box: for instance, confusing 'building' crops for 'tree' or 'sky' class.
- We did not mix *stuff* classes that contain the suffix '-other', as they usually contain a high variety of classes by themselves that are difficult to categorize. Exception: when the entire super-class (grouping of COCO-Stuff classes) can be mixed into one, including classes with suffix '-other'.
- Only merge classes that by individual inspection, are semantically close.

## C    METHODS UNDER STUDY

**G2im** (Ashual & Wolf, 2019) uses a scene graph as input conditioning, where nodes represent objects in the scene with their class annotation, and edges represent the relative positions among those objects. The scene graph is used to generate instance segmentation masks and instance bounding box layouts, which are fused to obtain a semantic segmentation mask of the scene. The resulting mask is then combined with object features, obtained separately by feeding object crops from training images to an auto-encoder, to create the final generated image.

**G2im-GT** (Ashual & Wolf, 2019) is a variant of G2im, where the instance segmentation masks and instance bounding box layouts are not generated, but directly taken from the ground-truth data.

**LostGANv1** (Sun & Wu, 2019) uses instance bounding box layouts with their corresponding class annotations as input conditioning. This input is used to compute object embeddings, which are subsequently used to generate instance segmentation masks. The instance segmentation masks are then fed to a scene generator which uses a spatial conditioning module, called ISLA-norm, that learns independent normalization parameters per object instance. These normalization parameters are applied through all stages of the scene generation process.

**LostGANv2** (Sun & Wu, 2020) takes the same input conditioning, and computes the same object embeddings as LostGANv1. In this case, both the instance segmentation mask generator and the scene generator are extended to follow a multi-stage generation process. In particular, the instance segmentation masks are refined using the mask from the previous stage and some intermediate image features from the scene generator. In turn, each time the mask is refined, it is used to update the normalization parameters used in the scene generator, therefore using a stage-wise ISLA-norm.

**OC-GAN** (Sylvain et al., 2020) takes the same input conditioning as the LostGANs. In contrast to LostGANv1's ISLA-norm, OC-GAN uses a variant of SPADE's normalization module (Park et al., 2019), which spatially conditions the scene generation on instance segmentation masks, per instance bounding box boundaries, as well as a semantic bounding box layout. Moreover, OC-GAN uses a scene-graph similarity module, called SGSM, that attends scene visual features with graph features, and matches global and local information from both the scene graph and the generated image to encode the location of specific objects in the scene. This scene graph similarity module is used at training time but discarded at test time.

## D    METRIC TABLES

Metrics reported in the analysis are detailed as follows: Table 2 presents the F1-score and accuracy, Table 3 the diversity score, and Table 4 the FID values for both scenes and objects.

## E    EXTENDED ABLATION METRICS

The variants of SPADE that are ablated are the following: (1) **SPADE** uses a global semantic segmentation mask and object boundaries as in (Park et al., 2019), (2) **SPADE(i)** that uses object embeddings and instance masks, (3) **SPADE(s)** that uses a global semantic segmentation mask obtained from fusing instance masks, and (4) **SPADE(oc)** that uses object embeddings, instance segmentation masks, object boundaries and a bounding box semantic layout, as proposed by Sylvain et al. (2020). The results of these comparisons are presented in detail in Tables 5, 6, 7, 8.

| | ↑ **Mean F1-score** | | | ↑ **Mean accuracy** | | |
|---|---|---|---|---|---|---|
| | $\mathcal{S}_s$ | $\mathcal{S}_u$ | $\mathcal{S}_{u^2}$ | $\mathcal{S}_s$ | $\mathcal{S}_u$ | $\mathcal{S}_{u^2}$ |
| GT | $75.2 \pm 0.0$ | $80.9 \pm 0.0$ | $64.7 \pm 0.0$ | $61.0 \pm 0.0$ | $70.1 \pm 0.0$ | $54.8 \pm 0.0$ |
| G2im* | $41.2 \pm 0.1$ | $48.1 \pm 0.5$ | $38.1 \pm 0.3$ | $12.6 \pm 0.1$ | $19.7 \pm 0.4$ | $9.7 \pm 0.3$ |
| G2im-GT* | $50.6 \pm 0.1$ | $57.5 \pm 0.7$ | $46.3 \pm 0.5$ | $24.2 \pm 0.1$ | $31.2 \pm 0.2$ | $19.8 \pm 0.2$ |
| LostGANv1 | $63.7 \pm 0.1$ | $73.8 \pm 0.6$ | $57.7 \pm 0.4$ | $\mathbf{30.9} \pm 0.1$ | $41.0 \pm 0.6$ | $26.8 \pm 0.3$ |
| LostGANv2 | $\mathbf{64.8} \pm 0.1$ | $75.9 \pm 0.1$ | $58.6 \pm 0.3$ | $30.8 \pm 0.1$ | $40.6 \pm 1.1$ | $26.3 \pm 0.5$ |
| OC-GAN | - | $80.0 \pm 0.1$ | $\mathbf{63.8} \pm 0.3$ | - | $49.6 \pm 0.9$ | $\mathbf{36.7} \pm 0.5$ |

Table 2: F1-score and accuracy averaged across examples in splits $\mathcal{S}_s$, $\mathcal{S}_u$ and $\mathcal{S}_{u^2}$. This table presents the mean and standard deviation of 5 different random seeds for the generative models. * Trained using the open-sourced code.

| | ↑ **DS** | | |
|---|---|---|---|
| | $\mathcal{S}_s$ | $\mathcal{S}_u$ | $\mathcal{S}_{u^2}$ |
| GT | - | - | - |
| G2im* | $0.56 \pm 0.06$ | $0.55 \pm 0.07$ | $0.56 \pm 0.06$ |
| G2im-GT* | $0.48 \pm 0.09$ | $0.47 \pm 0.11$ | $0.48 \pm 0.09$ |
| LostGANv1 | $0.47 \pm 0.09$ | $0.46 \pm 0.09$ | $0.49 \pm 0.08$ |
| LostGANv2 | $0.45 \pm 0.09$ | $0.43 \pm 0.09$ | $0.46 \pm 0.08$ |
| OC-GAN | - | $0.14 \pm 0.07$ | $0.13 \pm 0.06$ |

Table 3: Diversity score using LPIPS metric in all evaluation splits $\mathcal{S}_s$, $\mathcal{S}_u$, $\mathcal{S}_{u^2}$. * Trained using the open-sourced code.

## F    ADDITIONAL QUALITATIVE RESULTS

Figure 4 shows more qualitative results, for state-of-the-art methods and two of the best ablated pipelines.

## G    ADDITIONAL METRICS PER CLASS

Figure 5 shows the class-wise precision, recall and consistency for the top 10 most represented foreground classes in the training set. Comparing the metrics in $\mathcal{S}_u$ and $\mathcal{S}_{u^2}$ to the ones in $\mathcal{S}_s$, we observe that: **G2im** exhibits lower recall for *giraffe* and *zebra* in $\mathcal{S}_u$, while all other class-wise metrics are similar and equally low across splits, suggesting that the method generates poor quality objects for all splits, as seen in Sections 4.2, 4.3 and 4.4. **G2im-GT** exhibits higher precision and consistency for *car* class, with only lower precision for *cat* and lower recall for *zebra* and *giraffe*; however, we cannot observe lower metrics for the $\mathcal{S}_{u^2}$ split. These observations could be explained by, on one hand, the fact that metrics are still fairly low compared to LostGANs and OC-GAN to observe significant differences between splits; on the other hand, using ground-truth masks could positively impact generalization for unseen splits as seen in Section 4.5. **LostGANv1** generally achieves higher scores than G2im and G2im-GT, but shows some signs of lower generalization for some classes in both splits. In particular, in $\mathcal{S}_u$, classes *giraffe* and *zebra* exhibit lower recall, while in $\mathcal{S}_{u^2}$, *cat*, *dog* and *truck* show lower recall; as an outlier, giraffe scores higher recall. Finally, for **LostGANv2**, we can observe a decrease in $\mathcal{S}_u$, in terms of precision for the classes *cat* and *dog*, while *giraffe* and *airplane* exhibit lower recall. In $\mathcal{S}_{u^2}$, the classes *cat* and *zebra* exhibit lower precision, while *person*, *cat*, *dog* and *train* present lower recall; finally, *zebra* and *airplane* score lower consistency. This suggests that more classes suffer from lower metric scores in $\mathcal{S}_{u^2}$ than in $\mathcal{S}_u$ (in line with Section 4.4), even for well represented classes such as the top 10 studied classes.

Note that we cannot compare to metrics in $\mathcal{S}_s$ for OC-GAN, as the code is not publicly available nor are its pretrained models. The authors kindly shared their evaluation images in $\mathcal{S}_u$ and $\mathcal{S}_{u^2}$.

| | ↓ **FID** | | | ↓ **Object FID** | | |
|---|---|---|---|---|---|---|
| | $\mathcal{S}_s$ | $\mathcal{S}_u$ | $\mathcal{S}_{u^2}$ | $\mathcal{S}_s$ | $\mathcal{S}_u$ | $\mathcal{S}_{u^2}$ |
| GT | 0.0 | 0.0 | 0.0 | 0.0 | 0.0 | 0.0 |
| G2im* | $62.3 \pm 0.1$ | $137.6 \pm 1.6$ | $99.9 \pm 1.0$ | $56.2 \pm 0.1$ | $112.3 \pm 0.5$ | $80.8 \pm 0.6$ |
| G2im-GT* | $40.7 \pm 0.1$ | $112.9 \pm 1.9$ | $82.1 \pm 0.8$ | $31.0 \pm 0.5$ | $85.2 \pm 0.9$ | $55.8 \pm 0.4$ |
| LostGANv1 | $15.2 \pm 0.1$ | $84.2 \pm 1.0$ | $57.2 \pm 0.4$ | $20.3 \pm 0.2$ | $71.9 \pm 0.5$ | $45.2 \pm 0.2$ |
| LostGANv2 | $\mathbf{12.8} \pm 0.1$ | $\mathbf{80.0} \pm 0.4$ | $\mathbf{55.2} \pm 0.5$ | $\mathbf{15.5} \pm 0.6$ | $\mathbf{67.1} \pm 0.7$ | $\mathbf{40.9} \pm 0.3$ |
| OC-GAN | - | $85.8 \pm 0.5$ | $60.1 \pm 0.2$ | - | $85.8 \pm 0.6$ | $59.4 \pm 0.3$ |

Table 4: FID for each data split and model. As a reference, ground-truth images from each of the $\mathcal{D}_s$, $\mathcal{D}_u$ and $\mathcal{D}_{u^2}$ splits are used accordingly.Both splits have different number of points, so FID not comparable across splits. This table presents the mean and standard deviation of 5 different random seeds for the generative models.* Trained using the open-sourced code.

| | ↑ **Scene Precision** | | ↑ **Scene Recall** | | ↑ **Scene Consistency** | |
|---|---|---|---|---|---|---|
| | $\mathcal{S}_u$ | $\mathcal{S}_{u^2}$ | $\mathcal{S}_u$ | $\mathcal{S}_{u^2}$ | $\mathcal{S}_u$ | $\mathcal{S}_{u^2}$ |
| LostGANv2 | $89.1 \pm 1.4$ | $87.5 \pm 0.3$ | $\mathbf{63.9} \pm 2.0$ | $\mathbf{58.0} \pm 0.9$ | $61.1 \pm 0.9$ | $43.5 \pm 0.4$ |
| OC-GAN | $89.2 \pm 0.9$ | $86.9 \pm 0.6$ | - | - | $63.6 \pm 1.0$ | $45.8 \pm 0.5$ |
| ISLA | $89.7 \pm 0.9$ | $88.4 \pm 0.5$ | $\mathbf{65.2} \pm 1.1$ | $\mathbf{58.5} \pm 0.7$ | $63.6 \pm 1.2$ | $45.9 \pm 0.8$ |
| SPADE | $89.4 \pm 0.4$ | $\mathbf{91.1} \pm 0.3$ | $53.2 \pm 0.5$ | $49.2 \pm 0.5$ | $61.5 \pm 0.6$ | $46.0 \pm 0.3$ |
| SPADE(i) | $85.9 \pm 0.7$ | $85.9 \pm 0.5$ | $53.0 \pm 0.5$ | $46.3 \pm 0.8$ | $49.5 \pm 0.4$ | $37.7 \pm 0.7$ |
| SPADE(s) | $90.0 \pm 1.0$ | $86.3 \pm 0.7$ | $47.8 \pm 1.2$ | $44.0 \pm 0.7$ | $53.1 \pm 0.4$ | $38.4 \pm 0.6$ |
| SPADE(oc) | $90.9 \pm 0.5$ | $90.1 \pm 0.6$ | $57.0 \pm 0.5$ | $53.2 \pm 0.7$ | $65.7 \pm 0.7$ | $47.6 \pm 0.3$ |
| ISLA+M | $91.6 \pm 0.4$ | $91.7 \pm 0.5$ | $57.7 \pm 1.4$ | $55.3 \pm 0.6$ | $68.1 \pm 1.4$ | $53.1 \pm 0.5$ |
| ISLA+SG | $90.5 \pm 0.5$ | $88.4 \pm 0.7$ | $\mathbf{64.9} \pm 0.7$ | $\mathbf{59.1} \pm 0.4$ | $66.1 \pm 1.1$ | $47.2 \pm 0.5$ |
| ISLA+M+SG | $\mathbf{92.1} \pm 0.5$ | $\mathbf{91.8} \pm 0.4$ | $58.8 \pm 1.4$ | $54.8 \pm 0.6$ | $\mathbf{72.5} \pm 0.7$ | $\mathbf{55.9} \pm 0.6$ |

Table 5: Ablation results for all ablated pipelines, scene-wise precision, recall and coverage for the two splits $\mathcal{S}_u$ and $\mathcal{S}_{u^2}$. This table presents the mean and standard deviation of 5 random seeds that control the input noise.

All in all, class-wise metrics follow a trend similar to the class averaged object-wise metrics. More specifically, we observe a metric degradation in $\mathcal{S}_u$, which is further accentuated in $\mathcal{S}_{u^2}$, while results in $\mathcal{S}_s$ present better metrics, as can be seen for LostGANv2. However, there is no strong correlation between a specific class and the generalization capabilities of each model. Trends are better observed in Figures 1d, 1e and 1f, where the metrics are averaged across all foreground classes in the dataset.

| | ↑ **Object Precision** | | ↑ **Object Recall** | | ↑ **Object Consistency** | |
|---|---|---|---|---|---|---|
| | $\mathcal{S}_u$ | $\mathcal{S}_{u^2}$ | $\mathcal{S}_u$ | $\mathcal{S}_{u^2}$ | $\mathcal{S}_u$ | $\mathcal{S}_{u^2}$ |
| LostGANv2 | $69.6 \pm 0.3$ | $73.2 \pm 0.5$ | $30.2 \pm 1.2$ | $25.8 \pm 0.5$ | $40.9 \pm 0.7$ | $36.3 \pm 0.6$ |
| OC-GAN | $70.4 \pm 0.7$ | $70.9 \pm 0.8$ | - | - | $48.2 \pm 0.9$ | $42.1 \pm 0.2$ |
| ISLA | $\mathbf{73.1} \pm 0.5$ | $74.7 \pm 0.6$ | $28.4 \pm 0.8$ | $24.9 \pm 0.7$ | $45.9 \pm 0.9$ | $40.5 \pm 0.2$ |
| SPADE | $68.1 \pm 0.6$ | $70.0 \pm 0.5$ | $24.5 \pm 0.6$ | $20.4 \pm 0.4$ | $38.1 \pm 0.9$ | $32.5 \pm 0.5$ |
| SPADE(i) | $69.5 \pm 0.8$ | $71.2 \pm 1.0$ | $21.1 \pm 1.2$ | $19.5 \pm 0.6$ | $35.6 \pm 0.9$ | $32.4 \pm 0.4$ |
| SPADE(s) | $70.4 \pm 1.2$ | $72.0 \pm 0.6$ | $21.7 \pm 0.9$ | $18.9 \pm 0.6$ | $35.0 \pm 1.0$ | $31.8 \pm 0.6$ |
| SPADE(oc) | $70.2 \pm 0.9$ | $73.6 \pm 0.4$ | $26.6 \pm 1.1$ | $24.5 \pm 0.5$ | $41.5 \pm 0.7$ | $35.7 \pm 0.7$ |
| ISLA+M | $73.8 \pm 0.4$ | $\mathbf{77.7} \pm 0.3$ | $\mathbf{31.6} \pm 0.2$ | $\mathbf{28.1} \pm 0.8$ | $49.3 \pm 0.6$ | $43.4 \pm 0.9$ |
| ISLA+SG | $70.5 \pm 1.4$ | $74.2 \pm 1.0$ | $\mathbf{32.2} \pm 0.8$ | $\mathbf{28.3} \pm 0.5$ | $45.3 \pm 1.3$ | $42.7 \pm 1.1$ |
| ISLA+M+SG | $72.6 \pm 1.7$ | $74.3 \pm 0.7$ | $28.7 \pm 1.2$ | $25.0 \pm 0.4$ | $\mathbf{50.6} \pm 1.2$ | $\mathbf{43.7} \pm 0.7$ |

Table 6: Ablation results for all ablated pipelines. Metrics reported are object-wise precision, recall and coverage for the two splits $\mathcal{S}_u$ and $\mathcal{S}_{u^2}$.This table presents the mean and standard deviation of 5 random seeds that control the input noise.

| | ↑ **Mean F1-score** | | ↑ **Mean accuracy** | | ↑ **DS** | |
|---|---|---|---|---|---|---|
| | $\mathcal{S}_u$ | $\mathcal{S}_{u^2}$ | $\mathcal{S}_u$ | $\mathcal{S}_{u^2}$ | $\mathcal{S}_u$ | $\mathcal{S}_{u^2}$ |
| ISLA | $76.7 \pm 0.3$ | $59.6 \pm 0.2$ | $43.1 \pm 0.7$ | $30.7 \pm 0.3$ | $\mathbf{0.41} \pm 0.10$ | $\mathbf{0.44} \pm 0.09$ |
| SPADE | $74.1 \pm 0.4$ | $57.6 \pm 0.2$ | $39.8 \pm 0.6$ | $25.0 \pm 0.3$ | $0.01 \pm 0.01$ | $0.01 \pm 0.01$ |
| SPADE(i) | $64.9 \pm 0.5$ | $52.6 \pm 0.3$ | $32.1 \pm 0.6$ | $23.8 \pm 0.3$ | $\mathbf{0.26} \pm 0.16$ | $\mathbf{0.26} \pm 0.15$ |
| SPADE(s) | $65.9 \pm 0.4$ | $53.2 \pm 0.3$ | $29.5 \pm 0.7$ | $22.4 \pm 0.5$ | $0.15 \pm 0.13$ | $0.17 \pm 0.14$ |
| SPADE(oc) | $75.9 \pm 0.5$ | $59.2 \pm 0.3$ | $39.3 \pm 0.9$ | $26.4 \pm 1.0$ | $0.17 \pm 0.07$ | $0.20 \pm 0.07$ |
| ISLA+M | $77.0 \pm 0.3$ | $61.1 \pm 0.2$ | $45.4 \pm 0.3$ | $29.0 \pm 0.3$ | $0.12 \pm 0.08$ | $0.11 \pm 0.06$ |
| ISLA+SG | $78.9 \pm 0.1$ | $62.2 \pm 0.1$ | $42.1 \pm 0.6$ | $31.4 \pm 0.8$ | $\mathbf{0.41} \pm 0.10$ | $\mathbf{0.44} \pm 0.08$ |
| ISLA+M+SG | $\mathbf{80.8} \pm 0.4$ | $\mathbf{64.3} \pm 0.4$ | $\mathbf{49.3} \pm 1.0$ | $\mathbf{34.5} \pm 0.3$ | $0.18 \pm 0.07$ | $0.19 \pm 0.08$ |

Table 7: Ablation results for all ablated pipelines. F1-score, accuracy and DS for the two splits $\mathcal{S}_u$ and $\mathcal{S}_{u^2}$. This table presents the mean and standard deviation of 5 random seeds that control the input noise.

| | # Parameters | ↓ **FID** | | ↓ **Object FID** | |
|---|---|---|---|---|---|
| | Generator | $\mathcal{D}_u$ | $\mathcal{D}_{u^2}$ | $\mathcal{D}_u$ | $\mathcal{D}_{u^2}$ |
| ISLA | 40.5M | $78.5 \pm 0.3$ | $53.5 \pm 0.5$ | $66.1 \pm 0.3$ | $38.2 \pm 0.3$ |
| SPADE | 40.8M | $83.8 \pm 0.3$ | $62.2 \pm 0.2$ | $70.8 \pm 0.1$ | $44.9 \pm 0.9$ |
| SPADE(i) | 40.8M | $98.4 \pm 0.4$ | $64.2 \pm 0.4$ | $76.3 \pm 0.3$ | $44.5 \pm 0.2$ |
| SPADE(s) | 40.8M | $98.4 \pm 0.7$ | $64.5 \pm 0.3$ | $77.8 \pm 0.3$ | $44.1 \pm 0.2$ |
| SPADE(oc) | 41.7M | $78.7 \pm 0.5$ | $55.6 \pm 0.2$ | $67.6 \pm 0.5$ | $40.2 \pm 0.1$ |
| ISLA+M | 35.3M | $75.2 \pm 0.3$ | $52.3 \pm 0.1$ | $61.7 \pm 0.1$ | $36.0 \pm 0.2$ |
| ISLA+SG | 47.5M | $78.4 \pm 0.6$ | $52.1 \pm 0.2$ | $62.6 \pm 0.4$ | $34.2 \pm 0.1$ |
| ISLA+M+SG | 42.3M | $\mathbf{72.6} \pm 0.5$ | $\mathbf{50.2} \pm 0.2$ | $\mathbf{59.2} \pm 0.2$ | $\mathbf{33.2} \pm 0.1$ |

Table 8: Ablation results for all ablated pipelines. Number of parameters per generator for each of the pipelines, scene FID and object FID for the two splits $\mathcal{S}_u$ and $\mathcal{S}_{u^2}$. This table presents the mean and standard deviation of 5 random seeds that control the input noise.

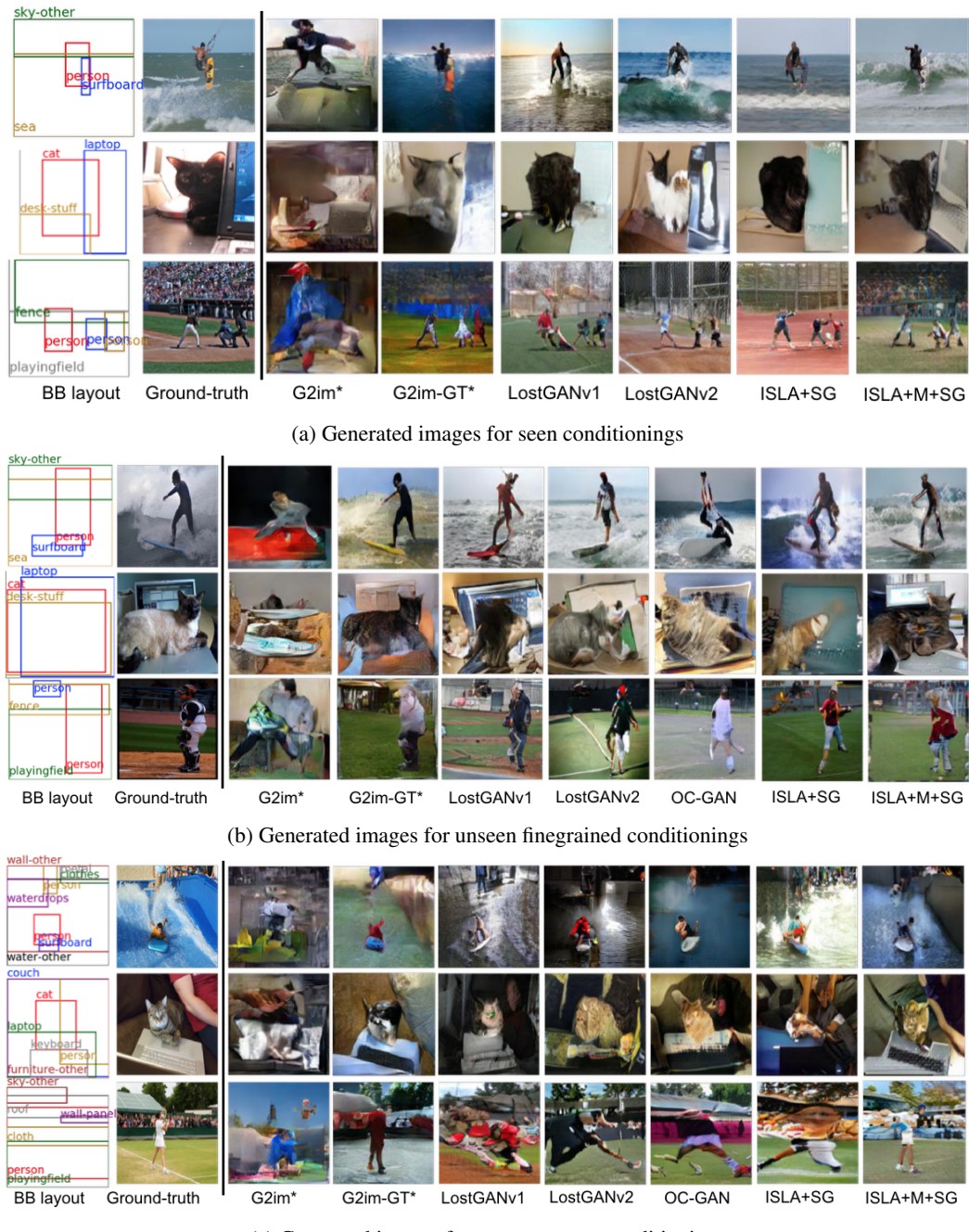

(a) Generated images for seen conditionings

(b) Generated images for unseen finegrained conditionings

(c) Generated images for unseen coarse conditionings

Figure 4: Extended qualitative results. Generated images with resolution $128 \times 128$ for several methods. * Trained using the open-sourced code.

## H    STATE-OF-THE-ART RESULTS FOR OUR BEST ABLATED METHOD

We consider our best model to be ISLA+SG, as stated in Section 4.5, to avoid leveraging ground-truth masks as in ISLA+SG+M and allow a fair comparison with LostGANv2 and OC-GAN, the two best existing methods as shown in Sections 4.2, 4.3 and 4.4.

In Table 9, we provide a direct comparison between LostGANv2, OC-GAN and our ISLA+SG in terms of: average scene precision, recall and consistency (also found with standard deviation and

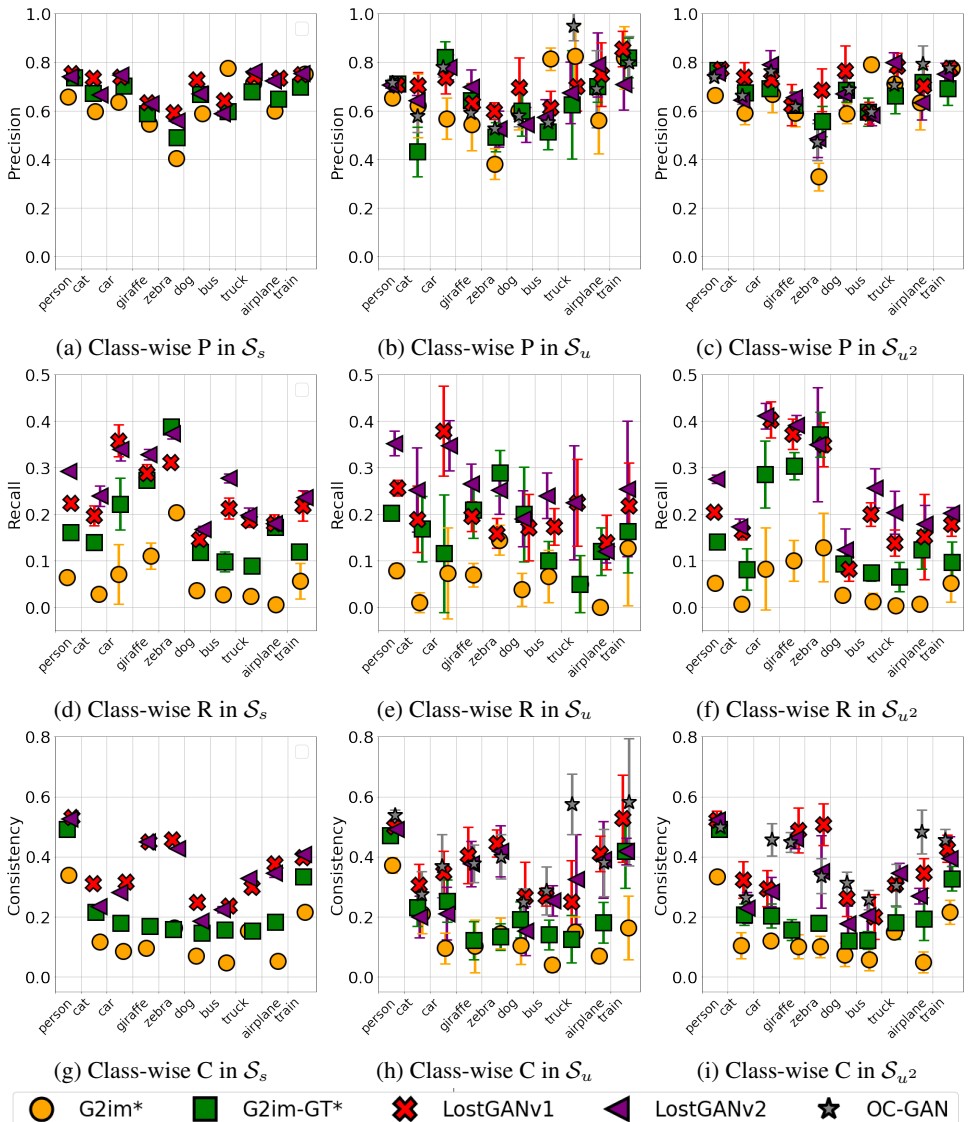

Figure 5: Class-wise comparison of state-of-the-art methods in terms of object-wise precision (P), recall (R) and consistency (C) (a,d,g) for seen conditionings ($\mathcal{S}_s$), (b,e,h) unseen finegrained conditionings ($\mathcal{S}_u$), (c,f,i) unseen coarse conditionings ($\mathcal{S}_{u^2}$). Plots show mean and std over 5 generation processes. *Trained using the open-sourced code. Best viewed in color.

for other ablated methods in Table 5), average object precision, recall and consistency (more details in Table 6), F1-score, accuracy and diversity scores (found in Tables 7 and 2) and finally, scene and object FID (in Tables 8 and 4).

As shown in the table, the method exploiting the findings of our analysis, namely ISLA+SG, achieves comparable or state-of-the-art results in the vast majority of image quality and diversity metrics, at the expense of exhibiting slightly worse performance in terms of consistency: F1-score and accuracy in both $\mathcal{S}_u$ and $\mathcal{S}_{u^2}$, and object consistency in $\mathcal{S}_u$.

# I    DETAILED INCONSISTENCIES BETWEEN SETUPS USED IN STUDIED METHODS

In this section we detail the main inconsistencies between the setups used to train and evaluate the studied methods in their original papers. The main inconsistencies are:

| | ↑SP | ↑SR | ↑SC | ↑OP | ↑OR | ↑OC | ↑F1 | ↑Acc. | ↑DS | ↓SFID | ↓OFID |
|---|---|---|---|---|---|---|---|---|---|---|---|
| *unseen finegrained conditionings $\mathcal{S}_u$* | | | | | | | | | | | |
| LostGANv2 (Sun & Wu, 2020) | 89.1 | 63.9 | 61.1 | 69.6 | 30.2 | 40.9 | 75.9 | 40.6 | **0.43** | 80.0 | 67.1 |
| OC-GAN (Sylvain et al., 2020) | 89.2 | - | 63.6 | 70.4 | - | **48.2** | **80.0** | **49.6** | 0.14 | 85.8 | 85.8 |
| ISLA | 89.7 | **65.2** | 63.6 | 73.1 | 28.4 | 45.9 | 76.7 | 43.1 | 0.41 | 78.5 | 66.1 |
| ISLA + SG | **90.5** | 64.9 | **66.1** | 70.5 | 32.2 | 45.3 | 78.9 | 42.1 | 0.41 | **78.4** | **62.6** |
| *unseen coarse conditionings $\mathcal{S}_{u^2}$* | | | | | | | | | | | |
| LostGANv2 (Sun & Wu, 2020) | 87.5 | 58.0 | 43.5 | 73.2 | 25.8 | 36.3 | 58.6 | 26.3 | **0.46** | 55.2 | 40.9 |
| OC-GAN (Sylvain et al., 2020) | 86.9 | - | 45.8 | 70.9 | - | 42.1 | **63.8** | **36.7** | 0.13 | 60.1 | 59.4 |
| ISLA | **88.4** | 58.5 | 45.9 | **74.7** | 24.9 | 40.5 | 59.6 | 30.7 | 0.44 | 53.5 | 38.2 |
| ISLA + SG | **88.4** | **59.1** | **47.2** | 74.2 | 28.3 | **42.7** | 62.2 | 31.4 | 0.44 | **52.1** | **34.2** |

Table 9: Direct comparison with the best state-of-the-art methods. We report average precision, recall and consistency for scenes (SP, SR, SC) and objects (OP, OR, OC), F1-score, Accuracy (Acc.), DS, scene and object FIDs (SFID and OFID). Results show the mean over 5 random seeds at test time. Note that *ISLA* is the same architecture as *LostGANv2*, but trained with the experimental setup proposed in the ablation, that leads to generally better results than the ones reported in the original paper (Sun & Wu, 2019). Bold numbers represent the best average metric across methods.

- **Different training sets**. Grid2im uses a training set of 25k images, while LostGANs uses 74777 images and OC-GAN uses 74121 images (according to communications with the authors). The discrepancy between LostGANv2 and OC-GAN comes from the way the data is filtered. All images are filtered to have between 3 and 8 objects, but LostGAN only considers objects that are segmented with polygons and not uncompressed RLE. This filtering, aside from altering the number of images used for training, can also alter which images are used and which objects are considered valid.

- **Different validation sets.** In Grid2im, they use only 2k images for validation, while Lost-GANs and OC-GAN use 3k images for reporting results.

- **Different reference distribution for FID**. When computing FID, LostGANs generated 5 samples per conditioning, ending up with 15k generated samples, against 3k real samples. In Grid2im and OC-GAN, they only generate 1 sample per conditioning, ending up with 3k generated samples. Note that changing the number of samples to compute FID, especially when using less than 10k samples, can significantly change FID scores (Kynkäänniemi et al., 2019; Xu et al., 2018), hindering the possibility of drawing robust conclusions when comparing methods.

- **Different image formats**. While Grid2im reported results computed on generated images saved in PNG format, LostGANs and OC-GAN do so on JPEG saved images. While this might seem a harmless difference, it can drastically affect the FID: computing FID on 2k validation coco-stuff images 128x128 using ground-truth PNG and JPEG images results in 20 points difference of FID score.

- **Training details.** Even though the studied methods do not use early stopping, they train for a different number of epochs. In the original papers, Grid2im and Grid2im GT have been trained for 82 epochs (on a much smaller dataset), while LostGANs have been trained for 125 epochs and OC-GAN for 170 epochs (on a much larger dataset). This gives more opportunity to OC-GAN to fit the data and potentially achieve better results. Note that in Section 4.5, training LostGANv2 and other ablation models for longer and using early stopping on FID improved results.

Additionally, each method uses different types of losses (one or many perceptual losses, using pixel-wise loss or not, type of adversarial loss, etc), and it is unclear whether any benefit comes from the loss choices, evaluation choices or model's contribution. Moreover, generator and discriminator choices affecting capacity and connectivity between modules change across papers.

