# OpenReview forum: "Generating unseen complex scenes: are we there yet?"
_ICLR.cc/2021/Conference — Reject_

### Official Review · AnonReviewer3 · 2020-10-23
**The paper provides a systematic evaluation of scene generation methods, but there are some concerns regarding novelty and the proposed setup.**

**Rating:** 6
**Confidence:** 5

**Review:**

Summary:

The paper provides a set of comparisons among different scene generation methods. It assesses ability of the models to fit the training set (seen conditionings), generalize to unseen conditionings of seen object combinations, and generalize to unseen conditionings composed of unseen object combinations. It finds that these models fit the training distribution with a moderate success, display decent generalization to unseen fine-grained conditionings, and have significant space for improvement when it comes to generating images from unseen coarse
conditionings.

############################################################

Strengths:

The authors provide a comprehensive set of experiments to compare performance of different scene generation methods using several metrics. This can be helpful for researchers to assess strengths and weaknesses of each model and its components, and helps them to gain insights into which aspect of models need to be improved.

###########################################################

Weaknesses:

1. While the comparisons among different scene generation methods are valuable, there are concerns about novelty of the paper especially since similar works have been published considering other computer vision tasks (e.g. [A, B]). The authors assess existing models in different settings and report their findings.
2. There are some concerns regarding the overall setup for the experiments. Out of the three cases considered, the ability of a model to fit its training set (case 1) is not very interesting practically as we are more interested in the model’s generalization. Case 3, generalizing to unseen conditionings composed of unseen object combinations, is also not expected as the models are not particularly trained to be generalizable to unseen coarse conditionings. If one is seeking models with generalization ability to unseen coarse conditionings, he/she needs to incorporate a form of transfer/meta-learning and train the models differently. Generalization to novel categories is also an issue in object-level GANs.
3. The paper claims that it is very hard to assess which models perform better due to “models being trained to fit different data splits, using different conditioning modalities and levels of supervision, and reporting inconsistent quantitative metrics (e.g. repeatedly computing previous methods’ results using different reference distributions, and/or using different image compression algorithms to store generated images), among other uncontrolled sources of variation”. However, I do not see a clear evidence supporting this. Each of the other papers (LostGAN, OC-GAN, etc.) provides a set of comparisons with other methods. The authors need to note specifically which setups are inconsistent among different papers. They report that LostGAN-v2 outperforms other models in most tasks. This is consistent with results reported in the LostGAN-v2 paper (although they evaluate their method on a smaller number of metrics).

##############################################################

Reason for rating:

While the paper provides a systematic evaluation of scene generation methods, there are some concerns regarding novelty and the proposed setup. I hope the authors clarify these in the rebuttal.

##############################################################

Additional comments:

There are some minor grammatical errors in the paper, and it needs further proofreading.

##############################################################

References:

[A] Are GANs Created Equal? A Large-Scale Study; Lucic et al.; NeurIPS 2018

[B] A metric learning reality check, Musgrave et al., ECCV 2020


################################################################

After author response: The authors have addressed my comment about inconsistent evaluation setups among different papers. However, I sill think novelty of the paper is limited as it is a conditional counterpart of [A]. As mentioned by other reviewers, findings of the paper are quite incremental and are in line with LostGAN-v2 although the authors use a more consistent evaluation setup.
Overall, I keep my current rating.

---

> ### Author Response · Authors · 2020-11-13
> **Answer to reviewer 4 [part 1]**
>
> We would like to thank all reviewers for their feedback and comments. As pointed out, the paper studies an important problem for the task of scene conditional image generation (R2), and the results, findings and insights provided by this work are valuable to audiences interested in its future developments (R4), and can be helpful to researchers to gain insights for future progress in this area (R3). Moreover, the paper provides reasonably thorough (R2), extensive (R4) and comprehensive (R3) experiments and evaluation results.
>
> We now address the reviewer's:
>
>
> **Similar works have been published.**
> ***We disagree on this premise, as we do not think conclusions from A or B apply in the task of conditional complex scene generation nor invalidate our work***. A focuses on the comparison of existing unconditional GANs for single-object generation, not on complex scene generation, and as such does not explore generalization in the way we do. B studies existing methods in metric learning, an unrelated task to complex scene generation. We acknowledge that there has been a plethora of analysis works in computer vision tasks. However, to the best of our knowledge we are the first ones to analyze complex scene generation pipelines. Thus, we argue that our analysis is novel.
>
> **Regarding interest in fitting the training data.**
> Looking at the training and validation set results is a standard practice in supervised machine learning approaches as ***it allows us to understand better model generalization*** and select proper models. By looking at the training and validation set results, one could detect overfitting and underfitting. The same methodology could be applied to complex scene generation pipelines. Thus, although it might not be interesting to look at the training set, we would argue that it is necessary to properly understand model generalization.
>
> **Not expected to be generalizable to unseen coarse conditionings.**
> ***All applications that motivate*** these conditional complex scene generation ***pipelines are expected to generalize to unseen object combinations***. In [1], the authors present an interactive scene generation tool, where a user defines input conditionings, which can be unseen at training time. It is worth highlighting how unlikely it is for a user to ask for exactly the same class combinations that were seen during training, given the number of possible classes (171) and their combinations into a 3 to 8 object scene (resulting in a huge number of possible class combinations). In [2], the authors motivate their work with an application that could generate scenes using under-represented or unseen characteristics (where unseen class combinations can be one of them) to augment a classification system with this generated data. Moreover, all existing pipelines are tested in datasets where the validation sets already contain unseen class combinations, as is the case for COCO-Stuff. Therefore, the models under analysis are indeed expected to generalize to unseen class combinations, as they encounter conditionings with unseen layouts and class combinations, even if they are not trained or tested specifically for it.
>
> **This answer continues in the next post.**
>
> References:
> *[A] Are GANs Created Equal? A Large-Scale Study; Lucic et al.; NeurIPS 2018*
>
> *[B] A metric learning reality check, Musgrave et al., ECCV 2020*
>
> *[1] Ashual, O., & Wolf, L. (2019). Specifying object attributes and relations in interactive scene generation. In Proceedings of the IEEE International Conference on Computer Vision (pp. 4561-4569).*
>
> *[2] Sun, W., & Wu, T. (2019). Image synthesis from reconfigurable layout and style. In Proceedings of the IEEE International Conference on Computer Vision (pp. 10531-10540).*

---

> ### Author Response · Authors · 2020-11-13
> **Answer to reviewer 4 [part 2]**
>
> **Details about inconsistent evaluation setups among different papers.**
> Thanks for this suggestion. We will include the exact differences in the evaluation protocols of different methods in the appendix of the paper. Below, we enumerate the main inconsistencies:
> * ***Different training sets***. Grid2im [1] uses a training set of ~25k images, while LostGANs uses 74777 images and OC-GAN uses 74121 images (according to communications with the authors). The discrepancy between LostGANv2 and OC-GAN comes from the way the data is filtered. All images are filtered to have between 3 and 8 objects, but LostGAN only considers objects that are segmented with polygons and not uncompressed RLE. This filtering, aside from altering the number of images used for training, can also alter which images are used and which objects are considered valid.
> * ***Different validation sets.*** In Grid2im, they use only 2k images for validation, while LostGANs and OC-GAN use 3k images for reporting results.
> * ***Different reference distribution for FID***.  When computing FID, LostGANs generated 5 samples per conditioning, ending up with 15k generated samples, against 3k real samples. In Grid2im and OC-GAN, they only generate 1 sample per conditioning, ending up with 3k generated samples.  Note that changing the number of samples to compute FID, especially when using less than 10k samples, can significantly change FID scores [3, 4], hindering the possibility of drawing robust conclusions when comparing methods.
> * ***Different image formats.*** While Grid2im reported results computed on generated images saved in PNG format, LostGANs and OC-GAN do so on JPEG saved images. While this might seem a harmless difference, it can drastically affect the FID: computing FID on 2k validation coco-stuff images 128x128 using ground-truth PNG and JPEG images results in ~20 points difference of FID score.
> * ***Training details.*** Even though the studied methods do not use early stopping, they train for a different number of epochs. In the original papers, Grid2im and Grid2im GT have been trained for 82 epochs (on a much smaller dataset), while LostGANs have been trained for 125 epochs and OC-GAN for 170 epochs (on a much larger dataset). This gives more opportunity to OC-GAN to fit the data and potentially achieve better results. Note that in Section 4.5, training LostGANv2 and other ablation models for longer and using early stopping on FID improved results. Additionally, each method uses different types of losses (one or many perceptual losses, using pixel-wise loss or not, type of adversarial loss, etc), and it is unclear whether any benefit comes from the loss choices, evaluation choices or model’s contribution. Moreover, generator and discriminator choices affecting capacity and connectivity between modules change across papers.
>
>
>
>
> References:
>
> *[1] Ashual, O., & Wolf, L. (2019). Specifying object attributes and relations in interactive scene generation. In Proceedings of the IEEE International Conference on Computer Vision (pp. 4561-4569).*
>
> *[2] Sun, W., & Wu, T. (2019). Image synthesis from reconfigurable layout and style. In Proceedings of the IEEE International Conference on Computer Vision (pp. 10531-10540).*
>
> *[3] Kynkäänniemi, T., Karras, T., Laine, S., Lehtinen, J., & Aila, T. (2019). Improved precision and recall metric for assessing generative models. In Advances in Neural Information Processing Systems (pp. 3927-3936).*
>
> *[4] Xu, Q., Huang, G., Yuan, Y., Guo, C., Sun, Y., Wu, F., & Weinberger, K. (2018). An empirical study on evaluation metrics of generative adversarial networks. arXiv preprint arXiv:1806.07755.*

---

### Official Review · AnonReviewer4 · 2020-10-28
**Official Blind Review #4**

**Rating:** 5
**Confidence:** 3

**Review:**

Summary:
This paper discusses the achievements and limitations of existing conditional generative models. As expected, while existing methods can generalize reasonably well to unseen layouts of seen object combinations, none of the methods can work well to unseen object combinations.

Strengths:
The paper provides extensive experiments to verify the performances of different methods under various conditions. The results should be valuable to audiences interested in future developments in the area.

Weaknesses:
There’s only one dataset that is being evaluated. Given that this paper is trying to test performances under various situations, it would be better to evaluate on more datasets to verify the results are not biased.

The conclusion is rather not surprising. I feel the paper would be stronger if it could provide more insight to the problems, especially on how to improve the generalization problem of existing methods. That is the interesting part and would really become a big contribution to the society. In the current form, it feels more like this paper only points out a problem that is already expected.

While the authors have identified some components that seem particularly useful in improving performances (Sec 4.5), I feel it is still not enough. In particular, most of the paragraphs are still simply spelling out the comparison results, without careful explanation of the insights. The authors should design more experiments to verify the hypotheses, explain the observed phenomena, and give details about the learned lessons. How these components are identified in the first place is worth mentioning too. Could there be other components that the authors are missing that could potentially be useful? Did the authors experiment with something else that turned out to be rather incremental? I think these are the questions that readers would want to know.

---

> ### Author Response · Authors · 2020-11-13
> **Answer to reviewer 4**
>
> We would like to thank all reviewers for their feedback and comments. As pointed out, the paper studies an important problem for the task of scene conditional image generation (R2), and the results, findings and insights provided by this work are valuable to audiences interested in its future developments (R4), and can be helpful to researchers to gain insights for future progress in this area (R3). Moreover, the paper provides reasonably thorough (R2), extensive (R4) and comprehensive (R3) experiments and evaluation results.
>
> We will now address the reviewer's concerns:
>
> **One dataset evaluated.**
> Although our analysis is performed on one dataset, we selected COCO-Stuff as it is very frequently used to evaluate complex scene generation and is a de facto benchmark for this task. In COCO-Stuff we can encounter the same problems as in other datasets: training data does not contain all possible class combinations and layouts, there is a long-tail class distribution and complex scenes.
>
> **Conclusions are not surprising.**
> The ***goal*** of this paper is not to end up with surprising findings, but rather to ***extract robust conclusions using thorough experiments and a fair experimental setup*** for all methods. Although the generalization problems may have been expected, we do believe that there is scientific value in confirming what is presumably expected, and can in turn help direct the attention of researchers in the field to questions that have long been seemingly overlooked. Moreover, to the best of our knowledge, our work is the first one to analyze in depth the complex scene generation pipelines.
>
> **Clarification to Section 4.5: Identification of pipeline components.**
> We identified the components based on the contributions proposed in each studied method: the usage of instance-wise ground-truth masks in Grid2im, ISLA spatial conditioning in LostGANv1, multi-stage mask refinement in LostGANv2 and, finally, SG module and spatial conditioning in OC-GAN. We will add this information to the paper.
> We did not specifically include the multi-stage mask refinement in the ablation, as it constitutes the difference between LostGANv1 and LostGANv2 and, as such, is already under study throughout the analysis sections (Sections 4.2, 4.3, 4.4). ***In the appendix E, we provide additional ablation results*** for other types of spatial conditioning that didn’t end up being as promising as the ones that made it into the main body.
>
> **How to improve the generalization of existing methods.**
> This ablation study in Section 4.5 enabled the identification of the most promising directions to improve generalization. Moreover, by exploiting our findings, we were able to show generalization improvements, and achieve state-of-the-art results in COCO-Stuff with the model denoted as ISLA+SG. In particular, we matched or surpassed all baselines in terms of scene and object Precision, Recall and Consistency (Tables 5 and 6), and obtained better FID for both $D_u$ and $D_{u^2}$ in scenes (\~2-3 points lower) and objects (\~4-6 points lower), compared to LostGANv2, the existing method with the lowest FID. We will make sure to highlight this in the paper. The findings of our studies can be summarized as follows:
>
> * ***Exploiting scene compositionality in the generator helps improve generalization***. In particular, we analyzed how different spatial conditioning modules affect the generation process, and observed that conditionings which explicitly take advantage of the compositionality of the scene, e.g. by considering instance segmentation masks,  promote generalization (ISLA > SPADE and its variants, as seen in Table 1 and Table 5 to 8 for extended results).
>
> * ***Semantically aware training losses promote generalization***. In particular, we analyzed the impact of using a scene-graph similarity (SG) loss, and observed that it leads to improved generalization performance (ISLA+SG vs ISLA). We hypothesize that this boost is due to computing the loss in the semantic space rather than in the pixel space.
>
> * ***Generating good quality masks remains one of the most promising directions to improve generalization***. Note the existing gap between models using ground-truth masks and those using generated masks (ISLA+M+SG vs ISLA+SG).
>
> * ***Improving the individual quality of the generated objects*** also remains an important direction to improve the generalization of conditional complex scene generation. This was observed throughout the whole analysis, where generated objects generally suffer from missing high frequency details. This is especially important given the very high quality results achieved by single object generation models.
>
> We will clarify and highlight the findings appropriately in the paper.

---

### Official Review · AnonReviewer2 · 2020-10-30
**[Official Review]**

**Rating:** 4
**Confidence:** 5

**Review:**

#### Summary ####
This paper studies the problem of scene conditional image generation with a focus on the evaluation of existing works towards unseen complex scene generation on the COCO-Stuff dataset. Specifically, it evaluates the model performances from three aspects, namely, image generation from seen conditionings, unseen fine-grained conditionings, and unseen coarse conditionings. For each evaluation, it computes the precision, recall, conditional consistency, F1-score, object accuracy, FID and diversity score for both object-wise and scene-wise measures.

#### Comments ####
This paper studies an important problem in scene conditional image generation. Reviewer appreciates the experimental efforts and reasonably thorough evaluations of existing methods. However, reviewer feels this is an okay submission but not good enough.

W1: Most of the “findings” described in the evaluations are either known or somewhat expected. Reviewer fails to obtain in-depth understandings and insights through reading the paper. Based on the evaluations, the proposed ablations studies (see Table 1) also failed to advance this field further. Though the paper contains several tables with ablation studies or side-by-side comparisons, it is hard to learn more from just the numbers in the table. Reviewer would highly suggest to either provide in-depth analysis (e.g., at category level, or at bounding box level) if you choose to go this route (see my points in W2) or propose a novel method that generalizes much better than previous work.

W2: The current analysis of scene conditional generation is very preliminary and limited. Reviewer would like to see the analysis per category or per bounding box (currently the performance evaluation is mixed). In addition, it would be good to visualize the learned feature maps of the existing work to see whether the model has achieved high-level understanding of the task or not, similar to what has been done before [Ref1-4]. This way, the quality and impact of the paper can be greatly improved, as readers will gain much better understanding from reading the paper than the current form.

[Ref1] GAN Dissection: Visualizing and Understanding Generative Adversarial Networks, Bau et al. In ICLR 2019.

[Ref2] Seeing What a GAN Cannot Generate, Bau et al. In ICCV 2019.

[Ref3] Understanding Neural Networks Through Deep Visualization, Yosinski et al. In ICML 2015 Deep Learning Workshop.

[Ref4] Visualizing and Understanding Convolutional Networks, Zeiler and Fergus. In ECCV 2014.

---

> ### Author Response · Authors · 2020-11-13
> **Answer to reviewer 2**
>
> We would like to thank all reviewers for their feedback and comments. As pointed out, the paper studies an important problem for the task of scene conditional image generation (R2), and the results, findings and insights provided by this work are valuable to audiences interested in its future developments (R4), and can be helpful to researchers to gain insights for future progress in this area (R3). Moreover, the paper provides reasonably thorough (R2), extensive (R4) and comprehensive (R3) experiments and evaluation results.
>
> We now address reviewer's concerns:
>
> **Improving generalization.**
> Although this is not the focus of our paper, we would like to point the reviewer to the improved results that we achieved by combining the most promising components of state-of-the-art models, which we identified in our ablation study. As a byproduct of our analysis, we were able to come up with model modifications that lead to ***superior generalization achieving state-of-the-art results*** with ISLA+SG (see Tables 1, 5, 6, 7 and 8 for direct comparison). In particular, when looking at Table 1, we outperform LostGANv2 (denoted as ISLA) in terms of e.g. FID for both generalization scenarios.
>
> **Findings known or expected.**
> To the best of our knowledge, our work is the first to analyze in depth the complex scene generation pipelines, and draw robust conclusions using thorough experiments and a fair experimental setup for all methods. We kindly ask the reviewer to provide us with references where this problem has been analyzed and studied and, which highlight the presumably known findings, as it can be very useful. As for the presumably expected findings, we do believe that there is scientific value in confirming what is presumably expected, and can help direct the attention of researchers in the field to questions that have long been seemingly overlooked.
>
> **Category or bounding box level analysis.**
> We would like to bring to the reviewer's attention that in our analysis we already provide an in depth analysis at both category and bounding box levels (see Figure 1 d-e-f), Figure 2 b-c) and Table 1). However, following the common practices of reporting results, we reported aggregates over classes or over bounding boxes. Moreover, we would like to emphasize that Figure 2 also shows that the classes in the long tail of the dataset exhibit lower results than those the more represented ones. Nevertheless, to address the reviewer’s concern we will add per class results of metrics to the appendix of the paper.
>
> **Visualizing feature maps.**
> The link between this experiment and our analysis is unclear to us. Could the reviewer give further information as to why this would enrich the understanding of generalization of conditional complex scene generation methods?

---

### Official Review · AnonReviewer1 · 2020-11-01
**Examination needs extensive nearest neighbors in the evaluation**

**Rating:** 4
**Confidence:** 4

**Review:**

Problem: There has been a plethora of work on image synthesis from a given layout of objects or label maps. However, it is not clear what has led to those results because there are no fixed backbone, optimization, training data, and evaluation protocol in each of them. This paper introduces a methodology to study three approaches (G2im, LostGAN, OC-GAN) that input a layout of objects to synthesize a new image.

What does the study include?: The study primarily assesses the ability of each model to (1) learn an effective mapping on the given data distribution; (2) generalize to unseen conditioning to seen object combinations; and (3) generalize to unseen conditioning to unseen object combinations.

Conclusions of the study: (1) The current approaches are able to nicely learn an effective mapping on the given data distribution and are able to generalize to unseen conditioning of seen object configurations. However, they struggle for unseen conditioning of unseen object configurations. (2) The authors further found three essential components that help in getting good performance. These are: (a) instance-wise spatial conditional normalization layer that increases the robustness of the model to unseen conditioning; (b)  scene-graph perceptual similarly helps improve scene generation; (c) improving the quality of label map leads to better results.


Setup of the study:

1. Data: There are three parts: (a) $D_{s}$: seen data on which the model is trained on; (b) $D_{u}$: unseen data in the validation set that has similar object combinations like $D_{s}$; and (c) $D_{u^2}$: unseen data in the validation set that does not have similar object combination like $D_{s}$. The authors use COCO-stuff dataset in the evaluation.

2. Evaluation Protocol: Following metrics are used in the study: (a) Precision; (b) Recall; (c) Consistency; (d) FID; (e) LPIPS-based Diversity Score (DS); and (f) object classification accuracy or image-to-set prediction F1 score.

Once you understand the setup, the analysis in Section-4 can be quickly understood. All the approaches follow a similar trend.


Major concerns with the study:

1. In my understanding (developed using the three-axis of study), one of the underlying goals is to contrast between memorization and generalization. *This study is incomplete without an extensive comparison with simple nearest neighbors*. The current model fairs well on $D_{s}$ that is a good sign about memorization. However,  $D_{u}$ is not too different from  $D_{s}$ in this case, especially when considering the COCO-stuff dataset. The examples shown in Figure-3 and Figure-4 can be very easily seen in the training set.

2. It is even more alarming to see that the approaches under study are not able to generalize to $D_{u^2}$. This is the scenario when we only change the object combinations.

3. Proper setup of nearest neighbors in both the input space and output space needs consideration. Without this, I find it hard to accept any claims about the different approaches.

4. The evaluation and the related work should also include Tan et al., CVPR 2019 (Text2Scene: Generating Compositional Scenes from Textual Descriptions).

Minor Point:

5. Section-4.5 (ground-truth masks): Though no example is provided in this paper, I looked through the generated masks in the original paper and it seems to be another issue that may be orthogonal to this study. I am not really sure how this point fits well to this study.

---

> ### Author Response · Authors · 2020-11-13
> **Answer to reviewer 1**
>
> We would like to thank all reviewers for their feedback and comments. As pointed out, the paper studies an important problem for the task of scene conditional image generation (R2), and the results, findings and insights provided by this work are valuable to audiences interested in its future developments (R4), and can be helpful to researchers to gain insights for future progress in this area (R3). Moreover, the paper provides reasonably thorough (R2), extensive (R4) and comprehensive (R3) experiments and evaluation results.
>
> We will now address the concerns:
>
> **Extensive comparison with simple nearest neighbor.**
> We would like to ask the reviewer if they could clarify what “proper setup of NN” means in this case. ***We already use nearest neighbor based metrics in our evaluation***. Note that Precision, Recall and Consistency are directly based on k-NN (where k=5) on the feature space.  Consistency provides information in the *input space* (conditionings) by assessing the overlap between the class set in the input conditioning that is used to generate an image, and the class set of the closest real sample. On the one hand, high consistency indicates the generated image depicts the input classes well, while low consistency means that the closest real image to the generated image contains a different set of classes. On the other hand, Precision and Recall provide information on the *output space* (generated images), to account for image quality and diversity.
>
> Moreover, we would like to highlight the differences between $D_u$ and $D_s$. Du contains unseen finegrained conditionings (unseen layouts) and $D_s$ contains seen conditionings, and so are different under the light of our generalization study. Although the object combinations of $D_s$ and $D_u$ are the same, their layouts are different. In particular, the size and position of the objects in the scene as well as the number of object instances per class are different. Although the specific examples in Figure 3 and 4 were chosen to ease the visual comparison between splits, all conditionings in $D_u$ are different from the ones of $D_s$ in at least one of the enumerated axes above. As an example, Figure 4, a) third row and Figure 4, b) third row show very different conditionings/ground truth images for the same set of classes.
>
> **Related work.**
> We also thank the reviewer for the provided reference, which we will include in the paper.
>
> **Generalization to $D_{u^2}$.**
> It is unclear to us why this point is a weakness, as we see this as an important observation of our study that can be surprising, but not alarming. In any case, ***this weakness exists in the SOTA models we studied, and not in our analysis***.
>
> **Ground-truth masks in the ablation.**
> The goal of this experiment was to analyze whether improving on the quality of the generated masks is a promising avenue for improving generalization (and how it compares to other possible improvement directions studied in the ablation). In this experiment, using ground-truth masks is a proxy for the ideal case, where generated masks have very high quality. The conclusions of this experiment showed that devoting efforts on improving generated mask quality could be as important as other possible directions to better the quality of generated scenes in different generalization scenarios.

---

### Author Response · Authors · 2020-11-24
**Updated paper**

Dear reviewers and AC,
we have updated the paper to include the feedback provided by the reviewers:
* rephrased findings of the analysis in the abstract and introduction
* added provided reference by R1
* added sentence at the end of each ablation paragraph summarizing the most relevant findings
* added sentence at the end of the ablation highlighting that as a byproduct of the analysis, we are able to get a new state of the art
* rephrased conclusions to highlight the main findings of the paper
* added content to the appendix as requested by the reviewers: (G) per class metrics and discussion, (H) direct comparison with state of the art methods, (I) inconsistencies among state of the art training and evaluation setups.

We hope that these clarifications are useful.

---

### Decision · Program_Chairs · 2021-01-07
**Final Decision**

**Decision:**

Reject

**Comment:**

The paper received mixed reviews that overall lean negative.

The main concern shared by reviewers is the novelty of the findings. Although the paper presents a systematic study that certainly has value, reviewers do not find sufficient insights from the analysis. The ACs agree with the reviewers that the paper is below the bar for acceptance.